# The Right-Handed Parallel β-Helix Topology of *Erwinia chrysanthemi* Pectin Methylesterase Is Intimately Associated with Both Sequential Folding and Resistance to High Pressure [note 1]

**DOI:** 10.3390/biom11081083

**Published:** 2021-07-22

**Authors:** Jessica Guillerm, Jean-Marie Frère, Filip Meersman, André Matagne

**Affiliations:** 1Centre for Protein Engineering, Laboratory of Enzymology and Protein Folding, InBioS, University of Liège, Building B6C, Quartier Agora, Allée du 6 Août, 13, 4000 Liège, Sart-Tilman, Belgium; jessica_guillerm@yahoo.fr (J.G.); jmfrere@uliege.be (J.-M.F.); 2Department of Chemistry, University of Antwerp, Groenenborgerlaan 171, 2020 Antwerp, Belgium; filip.meersman@uantwerpen.be

**Keywords:** protein folding, parallel β-helix, repeat proteins, circular dichroism, high pressure, kinetic intermediate, sequential pathway, contact order, dry molten globule

## Abstract

The complex topologies of large multi-domain globular proteins make the study of their folding and assembly particularly demanding. It is often characterized by complex kinetics and undesired side reactions, such as aggregation. The structural simplicity of tandem-repeat proteins, which are characterized by the repetition of a basic structural motif and are stabilized exclusively by sequentially localized contacts, has provided opportunities for dissecting their folding landscapes. In this study, we focus on the *Erwinia chrysanthemi* pectin methylesterase (342 residues), an all-β pectinolytic enzyme with a right-handed parallel β-helix structure. Chemicals and pressure were chosen as denaturants and a variety of optical techniques were used in conjunction with stopped-flow equipment to investigate the folding mechanism of the enzyme at 25 °C. Under equilibrium conditions, both chemical- and pressure-induced unfolding show two-state transitions, with average conformational stability (Δ*G*° = 35 ± 5 kJ·mol^−1^) but exceptionally high resistance to pressure (*P*_m_ = 800 ± 7 MPa). Stopped-flow kinetic experiments revealed a very rapid (τ < 1 ms) hydrophobic collapse accompanied by the formation of an extended secondary structure but did not reveal stable tertiary contacts. This is followed by three distinct cooperative phases and the significant population of two intermediate species. The kinetics followed by intrinsic fluorescence shows a lag phase, strongly indicating that these intermediates are productive species on a sequential folding pathway, for which we propose a plausible model. These combined data demonstrate that even a large repeat protein can fold in a highly cooperative manner.

## 1. Introduction

Proteins are synthesized on ribosomes, in the form of long amino acid chains, with the sequence coding for their three-dimensional structure [1,2]. Unraveling the details of the mechanisms by which a disordered polypeptide chain folds so rapidly to a specific and functional structure, not only in the test tube but also, and most remarkably, in the crowded environment of the cell [3,4], remains a fundamental challenge in modern structural biology. The refolding of proteins in vitro from inclusion bodies [5] for biotechnological and biomedical applications remains a major bottleneck in commercial and academic laboratories, and, moreover, the misfolding and aggregation of several proteins into toxic species is a hallmark of some of the most devastating human diseases [6]. In either situation, understanding what goes wrong in folding can be just as important as understanding how the right fold is achieved [7].

Most of the information available to date has been gathered using small soluble globular proteins (i.e., generally less than circa 100–150 residues), which often fold fast via two-state kinetic mechanisms [8,9,10,11,12,13,14], without a significant population of partially folded species, although complexities in some reactions have been observed [14,15,16,17,18,19,20,21]. Thus, despite the wide variety of native structures examined, common principles have emerged [22,23,24,25,26,27,28,29,30] and a unifying mechanism for protein folding has been proposed [31]. An important finding is that the folding rate of many proteins is inversely correlated to the average sequence distance between residues that form native contacts (known as “contact order”) in the native state [32,33]. Thus, the folding of large, multi-domain proteins, which are characterized by intricate topologies and numerous interactions between residues distant in the sequence (i.e., long-range interactions), is relatively slow in comparison with that of small, single-domain proteins. Although the high cooperativity of protein globular structures arises in part from long-range interactions, large proteins often fold and unfold through partially folded intermediate species. These are prone to aggregation and render the study of these proteins very difficult. However, proteins made up of more than 150 residues constitute the major fraction of all proteomes [4] and thus, there is a need for a better description of the stability and folding of medium-to-large-size globular proteins [34], and also of their self-assembly into macromolecular complexes [35]. 

As an alternative to globular proteins, repeat proteins [36] offer an attractive model to investigate the folding and stability of large proteins [37,38,39,40,41,42,43,44,45,46]. They show a distinctive modular nature, characterized by the succession of homologous structural motifs (termed repeats or coils), which stack up to form generally elongated, non-globular structures. In contrast with typical globular proteins, repeat proteins display the architectural simplicity of their repeats and are dominated by short-range interactions. In particular, all-β repeat proteins have not been characterized in as much detail as all-α and mixed α/β repeat proteins, and it will be interesting to compare them with globular proteins and see how they differ in their folding properties [45].

*Erwinia chrysanthemi* is responsible for soft-rot diseases in a wide range of plant species [47]. Pectin methylesterase (PemA) (EC 3.1.1.11) from *E. chrysanthemi* 3937 is a large monomeric enzyme of 342 residues (*M*_r_ 36953), which catalyzes the essential first step in the bacterial invasion, namely the deesterification of the methylated α-(1-4)-linked D-galacturonosyl residue component in the pectin molecule of the plant cell wall [48,49]. The X-ray structure of PemA [49,50] reveals a right-handed parallel β-helix fold (Figure 1), with a deep cleft on the surface of the enzyme, where the two conserved catalytic aspartate residues, at positions 178 and 199, are found. The polypeptide backbone folds into three parallel β-sheets, with the strands of successive repeats stacking on top of each other, which form the β-helix core of the protein, and several loops of variable lengths, resulting in a large elongated right-handed coil.

The right-handed parallel β-helix architecture is common in both secreted and membrane-bound microbial proteins that mediate diverse interactions with the extracellular medium. In particular, it has been discovered in bacterial, fungal, and viral adhesins, and also in various enzymes involved in the degradation and modification of carbohydrates, some as a virulence factor [36,50,51,52,53]. Furthermore, the naturally occurring right-handed parallel β-helix fold was proposed [54,55,56] as a plausible model to describe primordial amyloid fibril structure (for a comprehensive review on amyloid formation and structure, see [6]) but this was ruled out on the basis of X-ray fiber diffraction data [57].

According to the common nomenclature [58], the three β-strands that make each repeat of a parallel β-helix are called PB1, PB2, and PB3 (highlighted in Figure 1B), and the connecting regions (turns or loops) following the β-strands are termed T1, T2, and T3 (Figure 1D), respectively. As with other β-helical proteins (see e.g., the pectate lyase PelC [59], the P22 tailspike protein (TSP) [60], and P69 pertactin [61]), the structure of PemA [49,50] reveals that the β-helix itself is compact and mostly hydrophobic inside, while long flexible peripheral T1 and T3 loops are found towards the *C*-terminal end of the protein. PemA displays an α-helix at the N-terminal end of the β-helix, and capping structures are found at both extremities, which protect the hydrophobic core from the solvent and prevent oligomerization between the monomers [62]. It has a deep active-site cavity along the parallel β-helix, formed by the T3-PB1-T1 region that contains the most conserved residues in pectin methylesterases [49,50]. Finally, PemA shows the distinctive extensive side-chain stacking observed with all parallel β-helix proteins [50,52,53,58,59]. Thus, aromatic stacks of phenylalanine and tyrosine (Figure 1C,D) are found between β-strands of different repeats in the central part of the parallel β-helix domain, which most likely contribute to the correct folding and the conformational stability of the protein [53,63,64,65]. In addition, an external asparagine stack of three residues is present [50] and a disulfide bridge between Cys192 and Cys212 might occur [50,53]. Interactions between repeats have been shown to play a critical role in the stability and cooperativity of repeat proteins [42].

TSP [60] is a large trimeric protein that arises from the assembly of identical 666-residue-long polypeptide chains. The major part (residues 143–540) of the protein shows a typical parallel β-helix fold [60], whereas both the *N*- and *C*-terminal domains display anti-parallel β-sheets. TSP was the first protein for which in vivo partially folded intermediates could be identified [66], and in vivo and in vitro folding could be directly compared [67,68,69,70,71,72,73]. The folding of PelC [74,75,76] and pertactin [77,78], two other proteins with a classical β-helix structure [59,61], has also been studied in some detail. Thus, with both proteins, equilibrium folding was reported to deviate from a two-state model and kinetic analysis indicated a multistep process with a significant population of on-pathway partially folded species.

In this work, the chemical- and pressure-induced unfolding transitions of PemA were studied using a combination of spectroscopic techniques. This analysis provides an estimate of the unfolding free energy of the protein and reveals a remarkable resistance to high pressure. Furthermore, kinetic data obtained with the help of a stopped-flow instrument indicate sequential folding, with significant population of productive partially folded species. In particular, the fluorescence experiments provide support for the transient formation of at least two on-pathway kinetic species. This study brings further insights into the cooperativity of repeat-protein folding and the effect of native-state topology in protein folding as a whole. Moreover, it highlights structural aspects linked to the absence of long-range contacts and the simple topology of repeat proteins, which breaks the correlation between the rate of folding and the density of direct interactions between residues distant in sequence (i.e., contact order), as found for small globular proteins.

## 2. Materials and Methods

### 2.1. Chemicals 

Ultrapure guanidinium chloride (GdmCl), pectin from citrus peel (76280), acrylamide, 8-anilino-1-naphtalene-sulfonic acid (ANS), and bromocresol purple were purchased from Sigma Chemical Co. All other chemicals were reagent grade.

### 2.2. Molecular Biology 

Standard procedures for recombinant DNA technology were used as described by Sambrook and Russell [79]. The *pBCKSpemA* plasmid [49] was used as a template to amplify the *P**emA* gene coding for *Erwinia chrysanthemi* 3937 pectin methylesterase, with its own 24 amino acid residue signal sequence. The forward (pemANdeI; 5′-ATTCATATGTTAAAAACGATCTCTGGAACC-3′) and reverse (pemAXhoI; 5′-ATTCTCGAGCGTCAGGGTAATGTCGGCG-3′) primers contained sites for NdeI (CATATG) and XhoI (CTCGAG) restriction enzymes, respectively. A polymerase chain reaction (PCR) using Taq polymerase was performed for amplification, as follows: initial denaturation of the DNA at 94 °C for 4 min with subsequent amplification for 25 cycles of incubation at 94 °C for 1 min, 55 °C for 30 s, and 72 °C for 3 min; for the last cycle, incubation at 72 °C was extended to 7 min. The PCR product was purified and ligated to the pJET1.2 cloning vector (CloneJET PCR Cloning Kit, Fermentas). The presence of the insert within the pJET1.2 plasmid was checked by colony-PCR on randomly selected transformants, and plasmids from colonies carrying an insertion were amplified and extracted. Their sequences were determined by the Sanger method at the GIGA GenoTranscriptomics technology platform (Liège, Belgium).

To achieve a high level of expression, the *PemA* gene was cloned into the expression vector pET20b(+), containing the ampicillin-resistance gene, using the NdeI and XhoI restriction endonucleases. Finally, *Escherichia coli* DH5-α competent cells (Invitrogen, Paisley, UK) were transformed by the pET20bpemA vector and spread out on LB (lysogeny broth) agar plate with 100 µg·mL^−1^ of ampicillin (Sigma). Transformed *E. coli* DH5-α cells were grown on lysogeny broth (LB) medium containing 100 µg·mL^−^^1^ ampicillin. The recombinant plasmid was extracted and purified from DH5-α cells before use for protein expression.

### 2.3. Enzyme Expression and Purification

The recombinant PemA gene, cloned in a pET20b(+) plasmid, was transformed into *E. coli* BL21(DE3) cells (Novagen Inc., Madison, WI, USA). Cells were transformed with pET20bpemA plasmid and spread out on a LB agar plate with 100 µg·mL^−1^ of ampicillin and 0.2% of glucose (*w*/*v*) for 48 h at 28 °C. Transformed *E. coli* BL21(DE3) cells were grown overnight at 28 °C in 100 mL of yeast extract tryptone (YT) medium containing 100 µg·mL^−1^ of ampicillin and 0.2% of glucose (*w/v*). Following inoculation with 8 mL of an approximately 16 h preculture, the enzyme was expressed in 2 L bottles containing 500 mL of YT medium and 100 µg·mL^−1^ of ampicillin. The 6 L culture was grown at 28 °C, and cell development was followed by absorbance measurements at 600 nm. At an absorbance of circa 0.6, the temperature was fixed at 18 °C, and cell growth was allowed to proceed for an additional 12–14 h. The periplasmic proteins were then extracted by osmotic shock as described [80]. The supernatant containing the periplasmic proteins was filtered on a 0.22 µm membrane and dialyzed against 13 L of 20 mM MES, pH 6 (buffer A). The sample was then loaded onto a 25 mL SP Sepharose Fast Flow column (GE Healthcare), equilibrated in buffer A. PemA was eluted with a linear NaCl gradient (0–190 mM) in buffer A, and fractions containing the protein (identified by SDS-PAGE) were pooled together to be dialyzed in 20 mM HEPES, pH 7 (buffer B). A second purification step was performed using a 1 mL MonoS HR 5/5 column (GE Healthcare), equilibrated in buffer B. PemA was eluted with a linear NaCl gradient (0–63 mM) in buffer B, and the purified protein was finally dialyzed against a 50 mM phosphate sodium buffer, pH 7. The quality of the recombinantly produced PemA enzyme was assessed in line with the best practice recommendations established by the ARBRE-MOBIEU and P4EU networks (see https://arbre-mobieu.eu/guidelines-on-protein-quality-control/) (accessed on 11 May 2021). Thus, the purity of the sample was checked by SDS-PAGE and was found to be above 98%. The size homogeneity of the protein was assayed by size exclusion chromatography (24 mL Superdex75 HR10/30; [PemA] = 50 µM, i.e., 1.85 mg·mL^−1^), and no significant oligomerization (≤ 1% dimer and no larger oligomeric species) was observed. Identity was confirmed by both intact protein mass determination (*M*_r_ 36954 ± 4; ESI-Q-TOF Ultima, Micromass, Manchester, UK) and N-terminal amino acid sequencing (ATTYN; Applied Biosystems 476A protein sequencer). The final PemA preparation (circa 30 mg) was stored at −20 °C. The PemA concentration was determined through absorbance measurements performed at 280 nm in a 1 cm pathlength cell, on the basis of the calculated [81] extinction coefficient value (49975 M^−1^∙cm^−1^).

### 2.4. PemA Activity Assay

Enzymatic activity was measured at 25 °C by a colorimetric assay using bromocresol purple. This reagent is used as a pH indicator to follow the time-course of proton release during the deesterification of pectin from citrus fruit. The assay was performed in a 0.5 mL solution of 150 mM NaCl, pH 6, containing 0.75% pectin (*w*/*v*) and 37 µM bromocresol purple. A standard curve was established by the titration of the assay solution with 100 mM HCl, in order to correlate the change of absorbance measured at 590 nm with the amount of protons released. Enzymatic activity measurements were performed by monitoring the change in absorbance at 590 nm, following the addition of 4 µL of enzyme (0.3 − 1.5 µM, final concentration).

### 2.5. Circular Dichroism Measurements

Far-UV circular dichroism (CD) spectra were recorded with a Jasco J-810 spectropolarimeter at 20 °C, using a 1 mm pathlength quartz Suprasil cell (Hellma), with protein concentrations of circa 0.09 mg/mL. Five scans (10 nm/min, 1 nm bandwidth, 0.2 nm data pitch, and 4 s DIT) were averaged, base lines were subtracted, and no smoothing was applied. Data obtained with an applied high-tension voltage above 600 V were not considered. Data are presented as the molar residue ellipticity ([⊖]_MRW_) calculated using the molar concentration of protein and the number of residues. Secondary structure analysis was performed using the CDSSTR [82,83] algorithm provided in the DichroWeb analysis server [84,85].

### 2.6. Chemical-Induced Unfolding Transitions

Equilibrium folding was studied at 25 °C in a 50 mM sodium phosphate buffer, pH 7. With GdmCl, native or fully unfolded samples incubated at various final denaturant concentrations were allowed to unfold or refold, respectively, and to equilibrate for at least 9 days (under these conditions, equilibrium is reached throughout the transition; see the results). In contrast, with urea, only fully unfolded samples, incubated for 3 h in 8 M denaturant, were left to refold and equilibrate for 20 h. Under these conditions, equilibrium was also reached throughout the transition, and the risk of protein carbamylation was minimized. Unfolding and refolding transitions were obtained by monitoring the changes in intrinsic fluorescence emission (λ_ex_ = 280 nm; λ_em_ = 342 nm) and CD at 218 nm, using a Varian Cary Eclipse spectrofluorimeter and a Jasco J-810 spectropolarimeter, respectively, both equipped with a thermostatically controlled cell holder. With all samples, the data were corrected for the contribution of the solution (buffer + denaturant). Denaturant concentrations were determined from refractive index measurements [86], using an Atago R5000 hand refractometer. A protein concentration of 0.12 mg·mL^−1^ (3.3 µM) was used throughout.

ANS-bound fluorescence measurements were performed using the same samples, with excitation at 350 nm, and emission spectra recorded from 420 to 600 nm. The fluorescence spectra were corrected for the background fluorescence of ANS. The ANS concentration (determined from the molar extinction coefficient of 4950 M^−1^·cm^−1^ at 350 nm; Merck Index, Merck & Co., Whitehouse Station, NJ, USA) was 660 µM and hence, [ANS]/[PemA] = 200.

### 2.7. Quenching of Intrinsic Fluorescence by Acrylamide

All measurements were performed at 25 °C in a 50 mM sodium phosphate buffer, pH 7, using a protein concentration of 0.10 mg·mL ^−^^1^ (2.7 μM). Native (in the presence of 0.4 M GdmCl) and unfolded (in the presence of 4 M GdmCl) PemA was diluted with increasing concentrations of acrylamide (ranging from 0 to 250 mM), and fluorescence emission spectra were recorded from 310 to 440 nm, following excitation at 295 nm. The solvent accessibility of tryptophan residues was estimated according to the Stern–Volmer equation [87]:(1)$$\frac{F_{0}}{F}=1\;+K_{\mathrm{SV}}\left[Q\right]$$
where *F*_0_ and *F* are the fluorescence intensity of the protein in the absence and presence of acrylamide, respectively; *K*_sv_ is the Stern–Volmer quenching constant, and [*Q*] is the molar concentration of acrylamide. 

### 2.8. Pressure-Induced Unfolding

PemA was dialyzed three times against 50 mM TrisDCl, pD 7.6, and concentrated to 18 mg·mL^−1^ (0.49 mM). The sample was stored overnight at 25 °C to ensure that all solvent-accessible protons were exchanged for deuterons. Just before measurement, the sample was briefly centrifuged to remove any possible insoluble aggregated form of the protein, although no evidence was found for aggregation of the enzyme. Pressure scans were performed in a diamond anvil cell (Diacell Products, Leicester, UK) at 25 °C, and IR spectra were recorded using a Bruker IFS66 FTIR spectrometer (Karlsruhe, Germany) as described [88]. A smoothing of 17 points and baseline correction were applied to the spectra before data analysis. Secondary structure content of the protein was determined as described [89]. Fourier self-deconvolution was used before curve-fitting analysis. 

### 2.9. Kinetics of Unfolding and Refolding

All experiments were performed at 25 °C in a 50 mM sodium phosphate buffer, pH 7, using protein concentrations of 0.10 mg·mL^−1^ (2.7 μM) and 0.14 mg·mL^−1^ (3.7 μM) for fluorescence and CD measurements, respectively. Refolding reactions were initiated by a 10-fold dilution of PemA unfolded for circa 18 h in 4 M GdmCl (under these conditions PemA unfolds with τ ≈ 10 s), with the refolding buffer containing various GdmCl concentrations, yielding final concentrations in the 0.2 to 0.9 M range. Conversely, unfolding reactions were initiated by a 10-fold dilution of native PemA with the same buffer containing various amounts of GdmCl to yield final concentrations ranging from 2.5 to 4.5 M.

CD kinetics were monitored following the change in ellipticity at 218 nm in a 0.1 cm pathlength cell, whereas intrinsic fluorescence data were collected using excitation and emission wavelengths of 280 nm and 342 nm, respectively, in a 1 cm pathlength cell.

### 2.10. Stopped-Flow Experiments

All fast-mixing experiments were performed using a Bio-Logic (Claix, France) SFM-400 stopped-flow device, coupled with a MOS-450/AF-CD spectrophotometer, as described [90], with experimental dead times of ~3 ms and ~7 ms for fluorescence and CD measurements, respectively. For fluorescence quenching experiments, acrylamide was added into the refolding buffer to yield a final concentration of 81 mM and total emission fluorescence above 320 nm was measured following excitation at 295 nm. Binding of the fluorescent dye ANS was performed as described [90]. ANS was included in the refolding buffer, leading to final ANS concentration of 270 µM (thus, [ANS]/[PemA] = 100). In all experiments, 6000 data points were sampled over the time course of one experiment.

### 2.11. Thermodynamic Analysis

The thermodynamic parameters for chemical- and pressure induced unfolding were obtained on the basis of a two-state (N ⇌ U), according to [88,91], respectively.

### 2.12. Kinetic Analysis

Each kinetic trace resulted from the accumulation of approximately five and ten experiments for fluorescence and CD measurements, respectively. The resulting multiple data sets were fitted separately. These traces were analyzed according to a sum of exponential terms: (2)$$y_{t}=y_{\infty }+\sum A_{i}e^{-k_{i}t}$$
where *y*_t_ is an observable parameter (i.e., CD or fluorescence); *A_i_* and *k_i_* are the amplitude and the rate constant of the *i*^th^ phase, respectively; *t* is the time, and *y*_∞_ is the equilibrium value of the observed property. The rate constants were obtained by averaging the data sets, and errors were calculated as standard deviations throughout.

The dependence of the unfolding and folding rate constants on denaturant concentration was analyzed according to the following linear relationship [8,18,92,93,94]:(3)$$\ln\left(k_{obs}\right)=\ln\left(k_{f}^{H_{2}O}\;e{\;}^{\frac{-m_{kf}}{RT\cdot \left[denaturant\right]}}+k_{u}^{H_{2}O}e{\;}^{\frac{m_{ku}}{RT\cdot \left[denaturant\right]}}\right)$$
where *k_obs_* is the rate of unfolding or refolding measured at various GdmCl concentrations; *k_f_*^H_2_O^ and *k_u_*^H_2_O^ are the values for folding and unfolding, respectively, in the absence of a denaturant; and *m_k_f__*/*RT* and *m_k_u__*/*RT* are proportionality constants which describe the denaturant dependence. *R* is the gas constant, and *T* is the absolute temperature.

The difference in free energy between the folded and unfolded conformations was also calculated using kinetic data:(4)$$\Delta G^{0}{\left(H_{2}O\right)}_{NU}^{kinetic}=-RT\cdot \ln\left(\frac{k_{f}^{H_{2}O}}{k_{u}^{H_{2}O}}\right)$$

The programs Sigmaplot 9.0 (SPSS Inc., Chicago, IL, USA), Bio-Kine 32 V4.45 (Bio-Logic), and QtiPlot 0.9.97.10 (ProIndep Serv S.R.L., Craiova, Romania) were used for nonlinear least-squares analysis of the data.

### 2.13. Analysis of the Folding Kinetics

For a sequential four-state reaction (scheme (5)), there are three macroscopic rate constants (λ_1_, λ_2_, and λ_3_), which characterize the interconversion between the first and second, second and third, and third and fourth species, respectively:(5)$$\begin{array}{c} U \end{array}\begin{array}{c} \lambda_{1}\\ ⇋ \end{array}\begin{array}{c} I_{1} \end{array}\begin{array}{c} \lambda_{2}\\ ⇋ \end{array}\begin{array}{c} I_{2} \end{array}\begin{array}{c} \lambda_{3}\\ ⇋ \end{array}\begin{array}{c} N \end{array}$$

An analytical solution [95,96] can be found for the time-course of the different species, which is given by:(6)$$\frac{d\left[U\right]}{dt}=A_{0}e^{-\lambda_{1}t}$$
(7)$$\frac{d\left[I_{1}\right]}{dt}=A_{0}\lambda_{1}\frac{e^{-\lambda_{2}t}-e^{-\lambda_{1}t}}{\lambda_{1}-\lambda_{2}}\;$$
(8)$$\frac{d\left[I_{2}\right]}{dt}=-A_{0}\lambda_{1}\lambda_{2}\frac{\left(\lambda_{2}-\lambda_{3}\right)e^{-\lambda_{1}t}+\left(\lambda_{3}-\lambda_{1}\right)e^{-\lambda_{2}t}+\left(\lambda_{1}-\lambda_{2}\right)e^{-\lambda_{3}t}}{\left(\lambda_{1}-\lambda_{2}\right)\left(\lambda_{2}-\lambda_{3}\right)\left(\lambda_{3}-\lambda_{1}\right)}\;$$
(9)$$\frac{d\left[N\right]}{dt}=A_{0}{\left(1+\frac{\lambda_{2}\lambda_{3}\left(\lambda_{2}-\lambda_{3}\right)e^{-\lambda_{1}t}+\lambda_{1}\lambda_{3}\left(\lambda_{3}-\lambda_{1}\right)e^{-\lambda_{2}t}+\lambda_{1}\lambda_{2}\left(\lambda_{1}-\lambda_{2}\right)e^{-\lambda_{3}t}}{\left(\lambda_{1}-\lambda_{2}\right)\left(\lambda_{2}-\lambda_{3}\right)\left(\lambda_{3}-\lambda_{1}\right)}\right)}_{\;}$$

The length of the lag phase (*τ*_lag_, termed lag time), of any given species (e.g., I_2_), is given by the abscissa at the inflection point of the corresponding kinetic curve [95]. This parameter can be calculated by solving the following equation:(10)$$\frac{d^{2}\left[I_{2}\right]}{dt^{2}}=0$$

The lag time also corresponds to the time necessary for the previous species in the reaction (e.g., I_1_) to reaches its maximum concentration. This is given by:(11)$$\frac{d\left[I_{1}\right]}{dt}=0$$

Similarly, the lag time for N (i.e., the time needed for I_2_ to reach its maximum concentration) is given by:(12)$$\frac{d\left[I_{2}\right]}{dt}=0$$

## 3. Results

### 3.1. Chemical-Induced Unfolding

The changes in secondary and tertiary structural content were followed as a function of the GdmCl concentration by CD at 218 nm and by tryptophan fluorescence emission at 342 nm. The five tryptophan residues in PemA, all located in peripheral loops of the β-helix (Figure 1B), give rise to a single broad fluorescence emission band with a maximum at 342 nm (data not shown), which is consistent with the partial burial of the indole groups into the native structure [49,50] (see also acrylamide quenching experiments below). Upon the addition of 2.5 M GdmCl, the intensity decreased by about ~75%, accompanied by a red-shift of the maximum to 355 nm, indicating that stable tertiary contacts are lost, and the tryptophan side chains are exposed to the solvent. The far-UV CD spectrum (Figure 2A) of the native enzyme is typical for a protein with a high fraction of β-sheet structure, as indicated by a broad minimum in molar ellipticity centered at circa 218 nm [97]. Deconvolution using CDSSTR [82,83] revealed a β-strand content of 37 ± 1 %, in good agreement with the 38 ± 2 % found in the X-ray structure of PemA [49,50]. In 2.5 M GdmCl, the band centered at 218 nm is lost and the CD spectrum indicates the formation of an unordered structure (Figure 2A).

The full recovery of the optical and, most significantly, catalytic properties of PemA after a complete unfolding/refolding cycle demonstrated unambiguously that unfolding by GdmCl is fully reversible. For enzymatic measurements, we have developed a method based on a colorimetric assay, using a pH indicator (i.e., bromocresol purple, p*K*_a_ ~6.4) with a pH transition zone of 6.8 to 5.2 (purple to yellow). The specific activity of the enzyme was found to be (8 ± 1) × 10^3^ µmoles∙min^−1^∙mg^−1^. This assay can normally be used in this pH range for any reaction that leads to the release of protons. 

Both spectroscopic techniques (Figure 3) indicated that GdmCl induces a single cooperative transition between the native and unfolded states. Data revealed, however, that, following 20 h of GdmCl-induced unfolding and refolding, transition curves did not coincide (hysteresis) for GdmCl concentrations in the range from 0.75 to 2 M (Figure 3, insets), indicating that the samples had not yet reached equilibrium. Remarkably, unfolding proved to be extremely slow under these conditions, as it took about 9 days to reach equilibrium, while refolding was complete within 20 h.

Following circa 9 days of incubation in GdmCl (0 to 2.6 M), PemA was found to unfold with full thermodynamic reversibility, and the coincidence of the transition curves (Figure 3C) obtained by intrinsic fluorescence emission and far-UV CD measurements indicated that secondary and tertiary structures were destabilized concomitantly. The dye ANS binds preferentially to partially folded protein molecules with exposed hydrophobic patches [13,90,98,99,100,101,102,103]. We could not detect ANS binding in the transition region, confirming that PemA unfolds cooperatively in a single two-state transition (N ⇌ U), where only the native (N) and the unfolded (U) states are significantly populated, not any partially folded intermediates. Therefore, we used a simple two-state model to calculate the thermodynamic parameters for the PemA unfolding transition shown in Figure 3: ∆*G*°(H_2_O)_NU_ = 31 ± 2 kJ·mol^−1^, *m*_NU_ = −30 ± 2 kJ·mol^−1^·M^−1^ and *C*_m_ = 1.0 ± 0.1 M. Note that a ∆*G*°(H_2_O)_NU_ value identical within the error limits was obtained using urea as the denaturant (pH 7, 25°C; ∆*G*°(H_2_O)_NU_ = 39 ± 7 kJ·mol^−1^, *m*_NU_ = −14 ± 2 kJ·mol^−1^·M^−1^ and *C*_m_ = 2.9 ± 0.1 M; data not shown).

### 3.2. Pressure-Induced Unfolding

FTIR spectroscopy was used in combination with pressure-induced unfolding for measuring PemA stability. Very similar to other β-helix proteins [104], the FTIR spectrum of native PemA (Figure 2B) displays a maximum of the amide I’ band at 1634 cm^−1^, typical of β-sheet structure [105]. At 1100 MPa, the broadening of the amide I’ band and the displacement of its maximum to around 1647 cm^−^^1^ revealed the formation of an unordered structure [105]. Three parameters were considered in the analysis (Figure 4), which provide details about the structural modifications of the protein upon pressure-induced unfolding [88], i.e., the absorbance at a fixed wavenumber, the wavenumber corresponding to the absorbance maximum of the amide band, and the width of the band. Thus, as the pressure was raised up to 600 MPa (Figure 4) no significant change in the IR spectrum was observed. In this pressure range, only a limited decrease in the wavenumber of the band maximum occurred (Figure 4B), which probably results from the effect of compression of the hydrogen bonds [106] and also from H/D exchange of the internal hydrogens [107]. Above 600 MPa, a cooperative displacement of the band maximum towards higher wavenumber values (Figure 4B) took place, together with a broadening of the band (Figure 4C) and significant changes in intensity. In particular, the decrease in band intensity observed at 1634 cm^−1^ (Figure 4A) indicates the disappearance of the native sheet structure and the shift of the band maximum from 1634 to 1647 cm^−1^ is consistent with an increased number of unordered structures.

Although the three parameters appeared to change cooperatively upon pressure increase, suggesting a simple two-state unfolding process as observed in chemical unfolding experiments, FTIR spectra (Figure 2B) indicated that pressure-induced unfolding of the enzyme is not reversible. Indeed, following the release of the pressure, the spectrum of PemA did not shift back to its original position and remained rather broad, indicating that the enzyme was left largely unordered. Nevertheless, the apparently cooperative transition between N and U was tentatively analyzed according to a two-state model (Figure 4), yielding the apparent values for the thermodynamic parameters given in the legend to Figure 4. Remarkably, the apparent unfolding free energy (∆*G*°(H_2_O)_NU_ = 33 ± 4 kJ·mol^−^^1^) is identical within the error limits to that found using chemical denaturants (both GdmCl and urea), indicating that the pressure-induced unfolding transition between N and U can also be satisfactorily described according to a simple two-state equilibrium.

### 3.3. Fluorescence- and CD-Detected Folding Kinetics

The unfolding of the enzyme was monitored by fluorescence, in the presence of GdmCl concentrations in the range of 2.5 to 4.5 M. Following manual mixing, single exponential fluorescence decays were observed throughout and analyzed by using Equation (2) with *i* = 1. The refolding kinetics of PemA at 0.4 M GdmCl, monitored by intrinsic fluorescence and far-UV CD spectroscopy at pH 7.0, 25 °C, are shown in Figure 5. A sum of four and three exponential functions (Equation (2)), respectively, was fitted to the data, yielding the kinetic parameters in Table 1. Measurements of CD at 225 nm (Figure 5B) revealed that a substantial part (~60%) of the CD native signal is restored within the dead time of mixing (circa 7 ms). In the fluorescence measurements, however, a burst phase was not observed, indicating that the tryptophan emission did not change within the mixing dead time (circa 3 ms). Thus, these kinetic experiments suggest that a transient intermediate accumulates in the dead time of stopped-flow mixing, which shows a substantial amount of secondary structure, as indicated by the presence of ~60% of the CD signal at 225 nm but lacks stable tertiary contacts, as shown by the absence of changes in tryptophan fluorescence. After this burst phase, three kinetic phases (numbered 2 to 4) could be resolved with the two optical probes, which show identical time constants (see Table 1). The two slowest phases (i.e., 3 and 4) could also be measured reproducibly, by the two methods, following manual mixing (dead time ~10 s; time constants of 20 ± 5 s and 70 ± 20 s).

In contrast to far-UV CD measurements, which detect three visible kinetic phases only, an additional exponential term was necessary to fit the stopped-flow fluorescence data satisfactorily. This term corresponds to the fastest detectable process (phase 1; *τ*_1_ = 1.0 ± 0.3 s) and, remarkably, the sign of its amplitude (negative, see comment in the footnote to Table 1), which accounts for ~6% of the total signal change (Table 1), is opposite to that of the following phases (positive). This results in a lag phase in the refolding kinetics, which is clearly visible (Figure 5A, inset) due to the low and negative amplitude of the fluorescence change corresponding to this transition. This is consistent with the transient accumulation of an obligatory intermediate [108]. This species apparently lacks a specific fluorescence signature and hence cannot be distinguished from the unfolded state, but it results in a delay in the accumulation of the native or other highly fluorescent species [109,110].

The folding reaction was monitored by tryptophan fluorescence at various GdmCl concentrations and a plot (known as a chevron plot [94]) of the natural logarithm of the apparent rate constants for both folding and unfolding against the denaturant concentration is shown in Figure 6. Refolding of the enzyme at [GdmCl] < 1 M showed the complex kinetics described above, with at least 4 phases (see Table 1), whereas unfolding in the presence of GdmCl concentrations in the range of 2.5 to 4.5 M showed single exponential kinetics. At concentrations of denaturant ranging from 0.6 to 4.5 M, the unfolding and slowest phase-related refolding branches of the plot were found to be close to linear. Thus, at concentrations above 0.6 M GdmCl, Equation (3) was used to fit the data, yielding values of the kinetic parameters given in the legend to Figure 6. Extrapolation of the unfolding rate constant (*k_u_*) at 0 M GdmCl indicates that, in the absence of a denaturant, unfolding is remarkably slow (*k_u_*^H_2_O^ = 1.2∙10^−7^ s^−1^ i.e., *t*_1/2_ ≈ 67 days). Analysis of the data highlights the strong influence of the denaturant concentration on the unfolding rate constant and explains why circa 9 days were required to reach apparent equilibrium in the transition region, starting from native PemA. Thus, it can be calculated that it takes about 15 and 2.5 days in the presence of 0.8 and 1.3 M GdmCl, respectively, for the unfolding reaction to be 95% complete. Note that the time constant value for unfolding at 2.5 M GdmCl was as high as circa 23 min (i.e., *t*_1/2_ ≈ 16 min), and hence, no data were collected at lower concentrations due to the experimental complications inherent to such long time scales.

### 3.4. ANS Binding

The time course of ANS fluorescence intensity during PemA folding (Figure 5C) was measured by the inclusion of the dye in the refolding buffer, and data reveal that ~95 % of the ANS fluorescence enhancement (i.e., binding) occurred in the dead time of the stopped-flow experiment and that the maximum was reached after circa 0.5 s. This was followed by a three-exponential decay of the intensity, reflecting structural changes associated with the exclusion of the dye from the surface of the refolding protein. These three phases show time constants (Table 1) very similar to those measured by the changes in tryptophan fluorescence and in far-UV CD, indicating that the presence of the dye did not significantly affect the kinetics. Furthermore, the increase in ANS fluorescence was found to be independent of the ANS-to-protein concentration ratio ([ANS]/[PemA]) in the range from 50 to 200 (data not shown). 

### 3.5. Quenching of Fluorescence by Acrylamide

The exclusion of tryptophan residues from the solvent as the protein folds can be followed by monitoring intrinsic fluorescence in the presence of acrylamide as a fluorescence quencher [103,111]. The X-ray crystal structure of PemA [49,50] shows that the five tryptophan residues of the enzyme are partially buried into the native structure, with about 37, 28, 2, 55, and 10% solvent accessible area for residues at positions 107, 269, 303, 317, and 361 (calculated using the Naccess program [112]), respectively. This is consistent with the significant, although moderate, red-shift observed as the protein unfolds (from 342 to 355 nm, data not shown). The accessibility of fluorophores to collisional quenching is analyzed in the form of Stern–Volmer plots, as shown in Figure 7. Linear regression analysis of the data using Equation (1) yielded an average value of the Stern–Volmer constant (*K*_SV_) for the native state (5.0 ± 0.1 M^−1^) that is half that of the unfolded state (10.0 ± 0.2 M^−1^), in good agreement with significant exposure of tryptophan residues upon unfolding.

To investigate the change of solvent accessibility of the tryptophan residues during refolding, it was followed in the absence and in the presence of 81 mM acrylamide and the ratio (*F*_0_/*F*) of the fluorescence intensity in the absence (*F*_0_) and in the presence (*F*) of acrylamide was measured as a function of time, as shown in Figure 8. Data were analyzed according to a sum of three exponential functions (Equation (2)) and the resulting time constants were again consistent with those obtained using other spectroscopic probes (Table 1), indicating that all three methods monitor the same steps of folding. In the dead time of the rapid-mixing experiment, a large decrease in *F*_0_/*F* was observed, which reflects the sequestering of tryptophan residues from the solvent during the first circa 3 ms. This was followed by a significant increase in the *F*_0_/*F* ratio and thus, by an apparent re-exposure of the aromatic side chains, in two successive phases. Finally, the last phase shows a weak decrease in amplitude (see Table 1), probably due to a slight reduction in exposure of tryptophan indole groups as the protein adjusts to its native structure.

### 3.6. Slow Phases in PemA Folding

A possible reason for the occurrence of slow phases during protein folding, with time constants in the 10–100 s range [91,113,114], is the *cis-trans* isomerization of Xaa-Pro peptide bonds [115]. PemA contains 11 prolines, out of which 10 are located in the peripheral loops of the β-helix [49,50]; they are all in the *trans* configuration, and it is thus possible that the isomerization of non-native prolyl peptide bonds in the unfolded state determines the folding kinetics of the enzyme. In particular, the two slowest phases (i.e., 3 and 4) with time constants around 25 and 80 s could be associated with this phenomenon. In order to test this hypothesis, a classical double mixing experiment [116] was performed. In this experiment, the protein was fully unfolded in 6 M GdmCl at 25 °C for 5 to 10 s only. Under these conditions, this was sufficient for complete conformational unfolding of the enzyme (*τ* ~ 7 ms), but it was too short to allow significant isomerization of proline-containing peptide bonds, which thus remained in their native-like isomeric state. Under such conditions, the intrinsic fluorescence of the native PemA was recovered in two well-resolved kinetic phases, with amplitudes and time constants (circa 25 and 80 s, data not shown) identical to those obtained in single mixing experiments. Thus, the dramatic reduction of the time left for the protein to unfold had no influence on folding, suggesting that the two slow phases are not associated with proline isomerization. In addition, no effect was observed when folding was followed in the presence (5:1 PemA to PPI ratio in the final refolding mixture; data not shown) of either trigger factor (modified W151F mutant; [117]) or SlyD [118], two peptidyl prolyl isomerases (PPI; [115]), which normally accelerate the isomerization of prolyl peptide bonds. In conclusion, although both the prolyl bonds might just not be accessible to the PPIs tested here, and *trans-*to-*cis* isomerization in the unfolded state might be too fast for the double mixing experiment, all kinetic phases observed in PemA folding (Figure 5 and Figure 8) are likely to be associated with conformational refolding and not with the isomerization of incorrect *cis* forms of prolyl peptide bonds.

### 3.7. Analysis of the Folding Kinetics

The close similarity of the rate constants measured for the three phases (2 to 4, see Table 1) by the various probes suggests that they represent well-defined, cooperative folding transitions. The data are consistent with a sequential four-state model (scheme (5)) for folding, where λ_1_ to λ_3_ represent the macroscopic rate constants measured for phases 2 to 4. In this model, U represents the burst phase intermediate, considered as the ensemble of unfolded conformations under the refolding conditions and which already contains a significant fraction of secondary structure, whereas I_1_ and I_2_ corresponds to ensembles of transient, partially folded species, and N stands for the native state. Using the rate constants measured by tryptophan fluorescence, this model was used to simulate the time-courses of the various species (Figure 9). It shows a distinct lag phase in the formation of both I_2_ and N, occurring in the first ~20 and ~100 s of the reaction, respectively. Lag times of 9 and 44 s are calculated using Equations (11) and (12), for I_2_ and N, respectively. The experimental observation of a lag phase in the first few second of the fluorescence-detected reaction (Figure 5A) suggests that an intermediate is also significantly populated between the burst phase species (i.e., U in scheme (5)) and I_1_. This species is not directly detected because its fluorescence intensity is not significantly different from that of both the unfolded and burst phase species, but it results in a delay in the formation of the three following kinetic species (i.e., I_1_, I_2_, and N), which are characterized by a gradual enhancement of the fluorescence intensity of the enzyme until the native signal is reached.

## 4. Discussion

PemA displays a typical right-handed parallel β-helix fold (Figure 1) [49,50], with three parallel β-sheets (PB1, PB2 and PB3) forming the β-helix domain. This structural organization determines the spatial arrangement of the connecting peripheral loops (T1, T2, and T3), which can adopt their native conformation only after the β-sheets have formed. Furthermore, long flexible peripheral T1 and T3 loops, and an α-helix are located at the *C*- and *N*-terminal ends of the β-helix, respectively, which protect the hydrophobic interior from the solvent and prevent oligomerization. This topology causes special requirements for the folding of the polypeptide chain. Remarkably, the location of the five tryptophan residues of PemA (i.e., W107, W269, W303, W317, and W361) on peripheral loops (Figure 1B) might enable discriminating between the formation of the β-helix domain, monitored by far-UV CD, and the structural organization and docking of the lateral loops, monitored by intrinsic fluorescence measurements.

The conformational stability of PemA was probed by adding denaturants or by increasing pressure. GdmCl- and urea-induced unfolding was fully reversible as probed by intrinsic tryptophan fluorescence, far-UV CD and enzymatic activity measurements. 

The denaturant-induced equilibrium unfolding of PemA (Figure 3) is well described by a two-state transition (N ⇌ U) between the native (N) and unfolded (U) states, without partially folded species. Remarkably, circa 9 days at 25 °C were needed to reach equilibrium all through the transition, because the protein unfolded and refolded extremely slowly in the transition region. Extremely slow unfolding kinetics were also observed for two other β-helical proteins [74,77] and, by analogy with proteins from hyperthermophilic organisms [119,120], this is likely associated with the compact, hydrophobic core formed by their β-helix domain. 

Analysis of the data according to a two-state model yielded experimental cooperativity parameters (*m* values) for the unfolding transition (−30 ± 2 and −14 ± 2 kJ∙mol^−^∙M^−1^ in GdmCl and urea, respectively), which are consistent with those calculated (−33 and −15 kJ∙mol^−1^∙M^−1^, respectively) on the basis of the size of PemA [121]. The *m* value depends on the efficiency of the denaturant (*m*_GdmCl_ ≈ 2.3·*m*_urea_ [121]) and is proportional to the change in solvent exposed surface area upon unfolding. It is a good indicator for the two-state character of a transition, because hidden intermediates in a transition would decrease its apparent value substantially [121,122,123]. Thus, the coincidence between the unfolding curves obtained by fluorescence and CD measurements (Figure 3C), and the good agreement between calculated and experimentally determined *m* values indicate that the enzyme unfolds in a cooperative transition that involves the entire molecule. This allows the free energy change for unfolding (Δ*G*°_NU_ = 35 ± 5 kJ∙mol^−1^) to be calculated with confidence. The two-state unfolding of PemA by denaturants contrasts with the three-state transitions observed for pertactin [77] and for the plasmid-encoded toxin (Pet) from *E. coli* [124], two other β-helical proteins. For them, the population of a partially folded intermediate was observed at about 1.5 and 1.1 M GdmCl, respectively. Finally, for a large fragment (termed Bhx) of TSP (residues 109–544), which encompasses the entire β-helix domain, Seckler and collaborators [70,125] showed that its unfolding is a two-state process in urea, at low protein concentrations (10 µg∙mL^−1^, i.e., circa 0.2 µM), low ionic strength, and low temperature (10 °C), yielding (pH 7) Δ*G*°_NU_ = 32 kJ∙mol^−1^ and *m*_NU_ = 12.7 kJ∙mol^−1^∙M^−1^. In this case, however, the experimental *m* value was much lower than expected (19.8 kJ∙mol^−1^∙M^−1^) for a protein of this size, suggesting that the experimentally observed unfolding transitions involved only a part of the β-helix domain [70,125].

Thus, despite its large size and modular nature, PemA shows a remarkably high degree of cooperativity in equilibrium unfolding, as expressed in its close adherence to a simple two-state model. In comparison with PelC, which is very similar in size to PemA (353 and 342 residues, respectively), the data suggest a more homogeneous distribution of stabilities across the PemA β-helix length and also perhaps a higher stability of the repeat interfaces [46]. With pertactin, the larger size (i.e., 539 residues) and higher stability of the *C*-terminal half [77] of the protein probably explains the non-cooperative equilibrium unfolding. Finally, for Pet [124], the occurrence of at least two structural domains besides the right-handed β-helical *C*-terminal domain [126] probably explains the complex multi-state transition.

The cooperative transition of PemA is also clearly evident from pressure-induced unfolding experiments. The FTIR measurements revealed that very high pressures (>600 MPa) are needed to unfold the enzyme. Pressure-induced unfolding was not reversible, but a tentative two-state analysis gave a Δ*G*°_NU_ value of 33 ± 4 kJ·mol^−^^1^, which is identical within the error limits to that obtained from denaturant-induced unfolding. The apparent volume change for the unfolding of PemA (Δ*V* = −41 ± 5 mL·mol^−^^1^) is relatively small for its size (it corresponds to 0.15% of its molecular volume, i.e., 27336 mL∙mol^−1^, as calculated with the help of the Brugel software package [127]), and it is in the lower range compared to other globular proteins [128,129]. In comparison, pressure induced-unfolding studies on staphylococcal nuclease, a small monomeric protein of circa 17 kDa, revealed a significantly larger volume change (Δ*V* = −75 mL·mol^−^^1^) [130]. This contrasts, however, with the volume change for another (all α) repeat protein, Nank1-7, containing seven ankyrin sequence repeats and consisting of 248 amino acids, which was found to be −44 mL·mol^−1^ [131]. Here, a systematic analysis suggested that the volume change is mainly due to the changes in cavity volume rather than differential hydration. This study suggested that the major contributing factors to pressure effects on proteins are their imperfect internal packing and the occurrence of dry (i.e., with very low solvent occupancy) cavities in the folded state. Furthermore, it supports the view that there is a strong correlation between internal cavity volume and the volume change for protein unfolding [130,132]. In PemA, the side chain alignment of aliphatic and aromatic residues inside the hydrophobic β-helix core does not allow the presence of extended cavities. Analysis of the enzyme using the Brugel software [127] reveals that it contains seven major cavities, with volumes ranging from 7.7 to 14.8 mL·mol^−1^. This implies that the overall cavity volume is larger than the observed volume change upon folding. Clearly, relating this reaction volume solely to changes in cavities would be over simplistic. Nevertheless, PemA shows a higher stability towards pressure-induced unfolding (*P*_m_ = 800 ± 7 MPa) than many other proteins [128,133,134]. 

A combination of complementary spectroscopic probes was used to monitor the folding kinetics of PemA (Figure 5 and Figure 8). These experiments revealed that very fast changes occurred during the 3–7 ms dead time of stopped-flow mixing, followed by at least three resolved phases that were observed by all spectroscopic techniques. Within the dead time, the enzyme acquired one half (~56 %) of its native ellipticity at 225 nm (Figure 5B), accompanied by the formation of hydrophobic patches that were accessible for the dye ANS (Figure 5C). Tryptophan fluorescence did not change in the dead-time reaction (Figure 5A), but the access of indole side chains towards quenching by acrylamide was reduced. These observations are consistent with very fast (*τ* < 1 ms) hydrophobic collapse, a partial burial of tryptophan side chains, and substantial secondary structure formation. Stable native tertiary contacts had presumably not yet formed, however, as indicated by the absence of fluorescence changes during the dead time of mixing. These features are reminiscent of the molten globule type of intermediates that have been observed in the folding of many proteins [135,136,137,138]. A species with partial formation of the β-helix domain, involving the stack of phenylalanine residues (F168, F189 and F209—Figure 1C) in the core of the enzyme, might be a plausible model for the burst phase intermediate. This hypothesis is based on the data [64] showing that the burial of the internal side chains of stacked phenylalanine residues in the central region of the parallel β-helix domain of TSP is critical for its folding. Furthermore, it is also supported by the finding that ANS binding during the burst phase is not very important (i.e., ~1.5-fold increase) when compared to the typical > 5-fold increase observed with other proteins (see e.g., [100,101,102,103]). Similar results were obtained with pertactin [78], and these observations are interpreted here in terms of the early formation of a dry molten globule intermediate [139], with a partially structured β-helical domain. This species provides only low surface binding for the dye and hence, is apparently already quite dehydrated and probably offers little solvent access to the protein interior. Finally, the loops, although probably forming dynamic and fluctuating structures, show a significant degree of hydrophobic collapse, however, with burial of the five tryptophan side chains in a non-native environment, as indicated by the low degree of acrylamide quenching.

Following the kinetically unresolved changes in far-UV CD, ANS fluorescence, and quenching of intrinsic fluorescence by acrylamide, a further minor increase (~5%) in ANS fluorescence was observed (Figure 5C), which reaches a maximum at circa 1 s. The native optical signals were then regained in three visible, well-resolved kinetic phases (numbered 2 to 4 in Table 1), which were observed by all four optical probes. The various methods yielded very similar time constant values (i.e., 6 ± 1 s, 23 ± 2 s, and 80 ± 20 s) for these phases, which are all compatible with the *cis*-*trans* isomerization of Xaa-Pro peptide bonds [113,114,115]. PemA contains 11 prolines, but double jump experiments and refolding assays in the presence of two different PPIs did not provide evidence that prolyl isomerization is involved in the folding kinetics of the enzyme. These data, together with the observation of a lag phase (phase 1, Figure 5A) in the stopped-flow intrinsic fluorescence experiments, suggest that the kinetic changes observed in phases 2 to 4 represent well-defined cooperative conformational transitions. These can be described by a sequential folding pathway (scheme (5)), with the population of four species, i.e., U, I_1_, I_2_, and N, wherein I_1_ and I_2_ represent two obligatory, on-pathway intermediates. Using the macroscopic rate constants given in Table 1, the time-course of the four species was simulated (Figure 9) on the basis of scheme (5). This indicates that I_1_ and I_2_ reach a maximum amount (~60 and ~50%, respectively) after circa 9 and 44 s, respectively. These values are greater than the length of the lag phase (time constant of circa 1 s) observed in fluorescence experiments (Figure 5A), which therefore, suggests that an additional kinetic intermediate accumulates between the burst phase species and I_1_. It exhibits a low fluorescence intensity, similar to the phase burst intermediate, and therefore makes it possible to observe a lag in the appearance of the following species, which exhibit a greater fluorescence intensity. Very few claims (see [109,110]) of obligatory intermediates in protein folding are supported by the observation of an appropriate lag period ([108]), because favorable circumstances are required, namely the formation of an early intermediate species with spectroscopic properties identical or nearly identical to the starting species. The lag phase (also sometimes referred to as an induction period) is simply due to the accumulation of this spectroscopically silent intermediate before the substantial formation of subsequent species, with native-like spectroscopic characteristics. Clearly, the observation of a lag phase (Figure 5A) in the stopped-flow fluorescence experiments provides convincing evidence for the significant population of at least two productive (i.e., kinetically competent) intermediate species on the folding route between the denatured and native states (scheme (5) and Figure 10). Observation of this phase is possible here because the fluorescence properties of the native state, and of the two intermediate species I_1_ and I_2_, are markedly different from those of the unfolded and early intermediate species. The analysis of PemA folding, wherein structurally defined intermediates accumulate on a preferred pathway, is consistent with the classical macroscopic view [22,23] that folding occurs through a statistically predominant route. The distinct structural species that pave the way to the native state correspond to large molecular ensembles of rapidly interconverting molecules, which display significant common structural features and a measurable degree of cooperativity [24,140].

The restoration of the native optical signals (i.e., ellipticity, fluorescence emission intensity, and access to acrylamide), together with the release of the dye (ANS) in three exponential phases, can be interpreted as follows (illustrated in Figure 10). The first event (phase 2) leads to the regaining of a large percentage (~80%) of native ellipticity at 225 nm and a mere ~15% of native fluorescence intensity. This could reflect the formation of virtually the entire β-helix domain with the peripheral loops remaining in a non-native collapsed state (represented by I_1_ in Figure 10). A species (termed “native-like”) with similar structural features has been observed [75] in the early stage of PelC folding, suggesting a common intermediate on the folding pathway of these two right-handed parallel β-helix proteins. Following the population of I_1_, a second species (I_2_, Figure 10) accumulates (phase 3) before the formation of the native structure. Although it has recovered ~93% of the native CD signal at 225 nm, it shows only ~45% of the native fluorescence signal, indicating that it is missing a significant fraction of the native tertiary contacts. The unexpected increase in acrylamide quenching demonstrates, however, that a substantial change in the environment of one or more of the five tryptophan residues takes place during this phase. An attractive model would involve the docking of the tryptophan indole side chains close to their native orientations, particularly those at positions 107, 269, and 317 (see Figure 1). Thus, W107 and W317 display the two most solvent-exposed side chains, with ~37% and ~55% accessibility, respectively [112]. The increase in acrylamide quenching in phase 3 might be caused by the reexposure of these tryptophan residues during the final folding of the peripheral loops. In contrast, the large (~30%) increase in tryptophan fluorescence in this phase might originate from the burial of W269 in its final hydrophobic environment, close to the central PB1 β-sheet. W269 shows enhanced fluorescence probably because it resides in a hydrophobic environment and because the neighboring tyrosines Y158 and Y181 (Figure 1) transfer energy to W269. Finally, we speculate that the strong (~55%) increase in intrinsic fluorescence and ~7% change in ellipticity (phase 4), which characterize the final conformational adjustments to achieve the native structure, might be associated with the folding of the *C*-terminal loop (residues 345 to 359). The latter, which forms two short helices, packs anti-parallelly against the PB2 β-sheet of the parallel β-helix [49,50]. This results in the transfer of the indole side chain of W361 to a more hydrophobic environment, in close proximity to Y167 (Figure 1). As described for W269, the final positioning of W361 could result in a substantial increase in fluorescence intensity.

The chevron plot in Figure 6 highlights the complexity of the refolding kinetics of PemA, which contrasts with the apparent simplicity of unfolding. Thus, the absence of curvature in the unfolding limb of the chevron plot suggests a single cooperative unfolding, without intermediates, i.e., no sequential unraveling of the repeat array, repeat by repeat [45]. In refolding at low denaturant concentrations, only the rate of the slowest of the four exponential phases varies significantly with GdmCl concentration. This supports our analysis (schematically summarized in Figure 10) that, from the burst phase intermediate to I_2_, only minor local conformational rearrangements occur, without major changes in the surface area accessible to the denaturant. Strong changes in solvent accessibility occur during the last phase of folding and therefore, its rate shows strong denaturant dependence [141]. Between 0.6 and 4.5 M, the unfolding and refolding branches of the plot for the slowest phase (i.e., phase 4 in Table 1) were found to be linear. Thus, in this concentration range, Equation (3) was used to fit the data, yielding the kinetic rate constants given in the legend to Figure 6. Using the values of *k_f_*^H_2_O^ and *k_u_*^H_2_O^, Equation (4) allows calculation of a Δ*G*°_kin_ value of 39 ± 8 kJ·mol^−1^, which is identical within the error limits to the value (32 ± 3 kJ·mol^−1^) obtained from the equilibrium unfolding transitions.

Interestingly, the value of the folding rate constant (i.e., 0.8 s^−1^) extrapolated at 0 M GdmCl is about five orders of magnitude lower than that (circa 2∙10^5^ s^−1^) estimated on the basis of relative protein contact order [32,33] (calculated with the help of Baker Laboratory Tool http://depts.washington.edu/bakerpg/contact_order/, accessed on 11 May 2021). Other repeat proteins (see e.g., [78,142]) also fold with rates much lower than predicted based on their native state topology. As discussed in [42], this is due at least in part to the unusual local topology of all repeat proteins, which lack sequence-distant contacts. This lack and the simple topology of repeat proteins result in uniform and low contact order, and hence, the calculation based on contact order leads to folding rates that are orders of magnitude too high. Clearly, contact-order models, as developed for small globular proteins, are inadequate for repeat proteins [42].

## Figures and Tables

**Figure 1 biomolecules-11-01083-f001:**
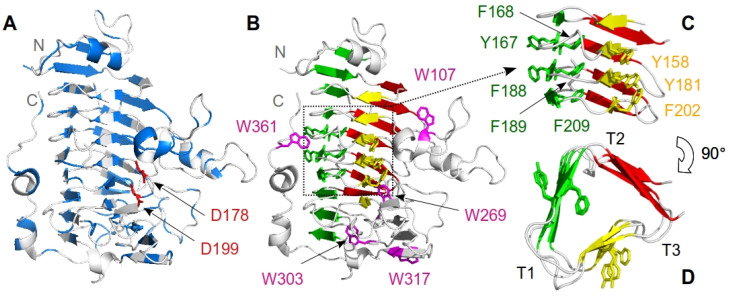
Schematic ribbon representation of the structure of *Erwinia chrysanthemi* 3937 pectin methylesterase (2NSP, [49]). The secondary structure elements are shown with (**A**) the hydrophobic residues in blue and the catalytic aspartate residues in red, (**B**) the three β-sheets in yellow (PB1), green (PB2), and red (PB3), and the tryptophan residues in purple. A closer view on the aromatic stacks in the central part of the β-helix is shown from (**C**) side and (**D**) top views. N and C indicate the *N*- and *C*-terminal ends of the polypeptide chain, respectively, and T1, T2, and T3 correspond to the turns following the three β-sheets. The figure was drawn using the open-source molecular graphics system PyMOL.

**Figure 2 biomolecules-11-01083-f002:**
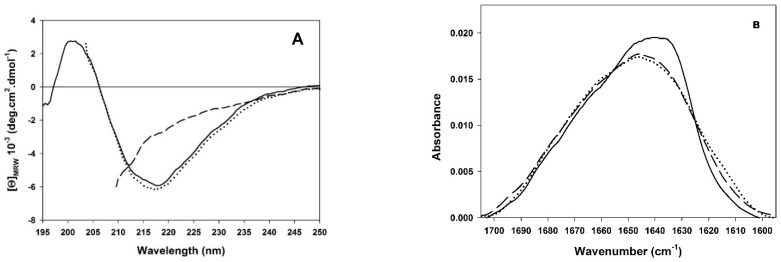
(**A**) Far-UV CD spectra of native (continuous line), unfolded in 2.5 M GdmCl (broken line) and refolded from 2.5 to 0.25 M GdmCl (dotted line) PemA. (**B**) FTIR spectra of PemA under native (33 MPa; continuous line), unfolding (1100 MPa; broken line), and refolding (i.e., 41 MPa after decompression; dotted line) conditions, with data normalized using the area under the amide I’ band equal to 1.

**Figure 3 biomolecules-11-01083-f003:**
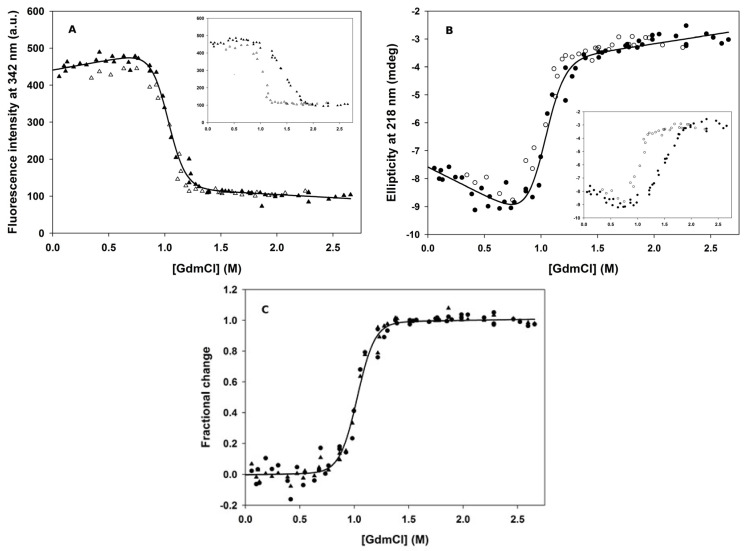
GdmCl-induced equilibrium unfolding transitions of PemA at pH 7 and 25 °C, monitored by (**A**) the change in fluorescence intensity at 342 nm and (**B**) the change in ellipticity at 218 nm. Closed symbols represent the addition of a denaturant to a solution of native protein, whereas open symbols are for the dilution of unfolded protein (in 3 M GdmCl) to the indicated concentrations of GdmCl. The samples were incubated at the indicated GdmCl concentrations for 9 days before the measurements. Data were analyzed on the basis of a two-state model [91], and the lines were drawn using (**A**) ∆*G*°(H_2_O)_NU_ = 32 ± 3 kJ·mol^−1^, *m*_NU_ = −31 ± 3 kJ·mol^−1^·M^−1^ and *C*_m_ = 1.0 ± 0.1 M, and (**B**) ∆*G*°(H_2_O)_NU_ = 29 ± 4 kJ·mol^−1^, *m*_NU_ = −28 ± 4 kJ·mol^−1^·M^−1^ and *C*_m_ = 1.0 ± 0.1 M. (**C**) Fractional change in signal [91] as a function of GdmCl concentration. Triangles and circles represent fluorescence and far-UV CD data, respectively. The line was drawn using the average value for the parameters obtained in fluorescence and CD experiments (see text). Insets in (**A**,**B**) represent data obtained after circa 20 h of incubation (see text).

**Figure 4 biomolecules-11-01083-f004:**
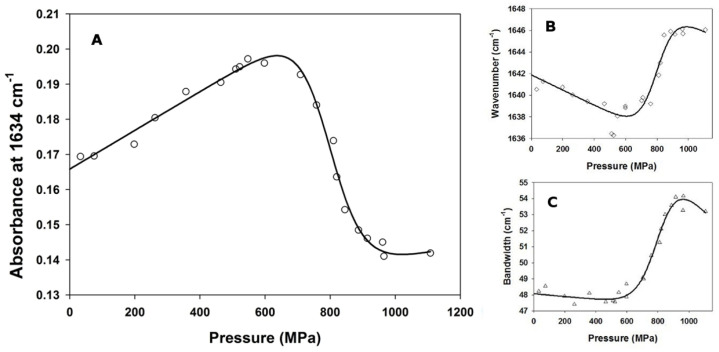
Pressure-induced unfolding curves of PemA followed by FTIR spectroscopy at 25 °C, through monitoring (**A**) the absorbance at 1634 cm^−1^, (**B**) the wavenumber corresponding to the band maximum and (**C**) the amide I’ bandwidth. The solid lines represent the best fits of Equation (3) in [88] to the data, yielding the following average apparent values for the parameters: ∆*G*°(H_2_O)_NU_ = 33 ± 4 kJ·mol^−1^, Δ*V* = −41 ± 5 mL·mol^−1^ and *P*_m_ = 800 ± 7 MPa.

**Figure 5 biomolecules-11-01083-f005:**
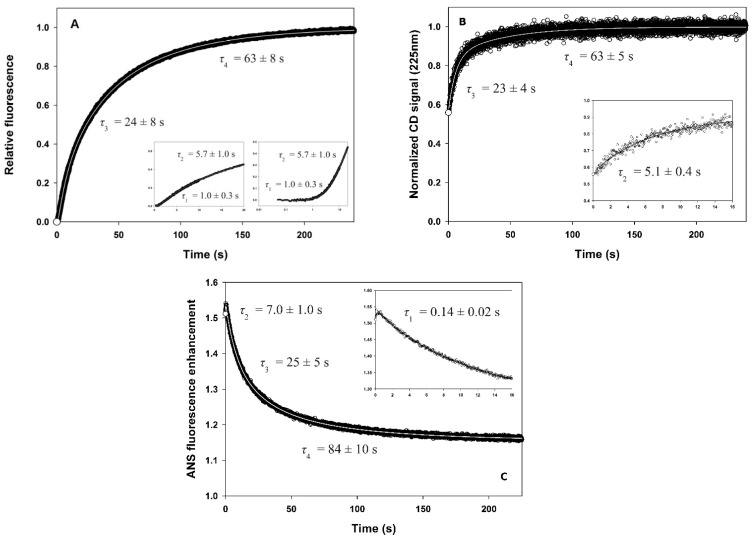
Refolding kinetics of PemA at pH 7, 25 °C, in 0.4 M GdmCl and 50 mM sodium phosphate, monitored by (**A**) total intrinsic fluorescence emission above 320 nm (after excitation at 295 nm), (**B**) CD at 225 nm, and (**C**) total ANS fluorescence emission above 450 nm. A sum of four and three exponential functions (Equation (2)) has been fitted to the fluorescence (both **A**,**C**) and CD data, respectively. Intrinsic fluorescence (**A**) and CD (**B**) data have been normalized to the total signal difference between the native (1) and unfolded (0) proteins under refolding conditions, whereas extrinsic fluorescence intensities in ANS binding kinetics (**C**) are expressed relative to the fluorescence of ANS in the presence of the unfolded protein in 4 M GdmCl. The resulting time constants are indicated. (○) Signal extrapolated to zero time from the kinetic data. Insets show the first 16 s of the reaction (note that intrinsic fluorescence data are displayed with both a linear and a logarithmic time scale).

**Figure 6 biomolecules-11-01083-f006:**
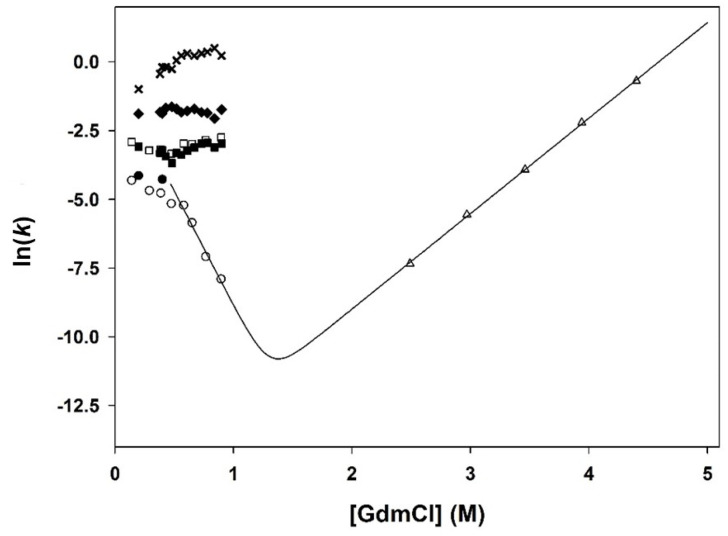
GdmCl concentration dependence of the rate constants for PemA folding at pH 7, 25 °C. Circles and squares represent the logarithm of the observed rate constants (in s^−1^) for slow refolding (phases 3 and 4 in Table 1); diamonds and crosses are for fast refolding (phases 1 and 2), whereas triangles correspond to the unfolding rate constant. Data obtained after rapid mixing are represented by closed symbols, while those obtained following manual mixing are shown by open symbols. The solid line represents the fit of Equation (3) to the data obtained with GdmCl concentration in the range from 2.5 to 4.5 M (triangles), using *k_f_*^H_2_O^ = 0.8 ± 0.2 s^−1^, *m_k_f__* = 21 ± 2 kJ·mol^−1^ ·M^−1^, *k_u_*^H_2_O^ = (1.2 ± 0.1) · 10^−7^ s^−1^, and *m_k_u__* = 9 ± 1 kJ·mol^−1^ ·M^−1^.

**Figure 7 biomolecules-11-01083-f007:**
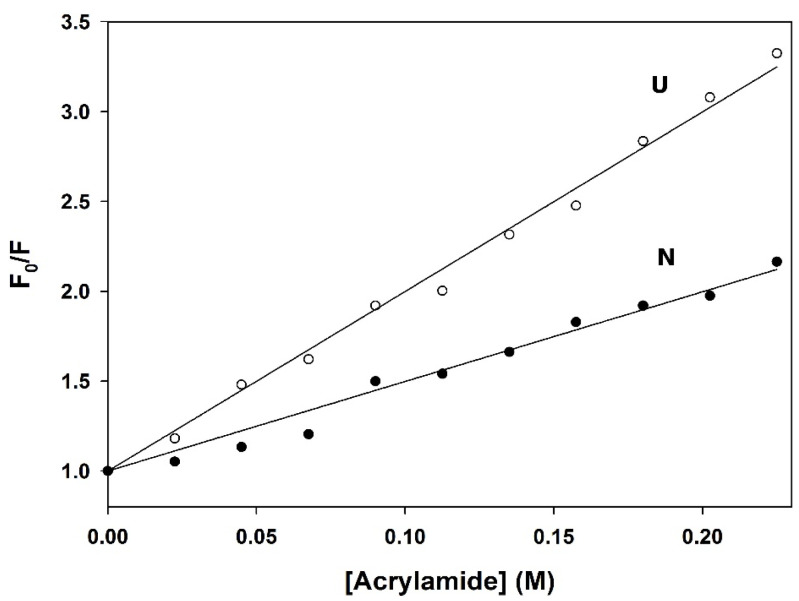
Stern–Volmer plots of *F*_0_/*F* versus quencher concentration for the native state (N) of PemA in 0.4 M GdmCl (●) and the unfolded state (U) in 4 M GdmCl (○). *F*_0_ and *F* are the fluorescence intensities in the absence and presence of acrylamide, respectively. Values of the average Stern–Volmer constant (*K_SV_*), i.e., 5.0 ± 0.1 M^−1^ (N) and 10.0 ± 0.2 M^−1^ (U), are given by the slope of the straight line fitted to the data according to Equation (1).

**Figure 8 biomolecules-11-01083-f008:**
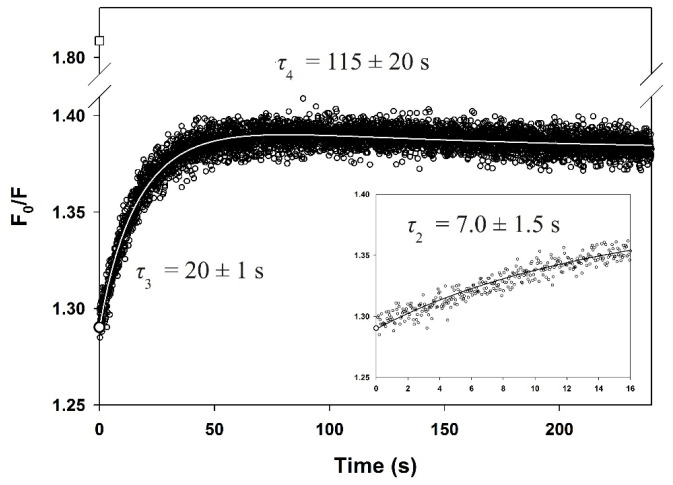
The quenching of intrinsic fluorescence by 81 mM acrylamide during the refolding of PemA. Refolding was carried out in 0.4 M GdmCl and 50 mM sodium phosphate at pH 7 and 25 °C. The data are plotted as the ratio of the intrinsic fluorescence in absence of acrylamide (*F*_0_) to that in the presence of the quencher (*F*). (□) is *F*_0_/*F* for the unfolded protein measured as the ratio of the fluorescence intensities of the protein in 4 M GdmCl, with and without acrylamide; (○) represents the signal of *F*_0_/*F* extrapolated to zero time, and the solid line represents the fit of a sum of three exponentials (Equation (2)) to the data. The resulting time constants are indicated, and the inset shows the first 16 s of the reaction.

**Figure 9 biomolecules-11-01083-f009:**
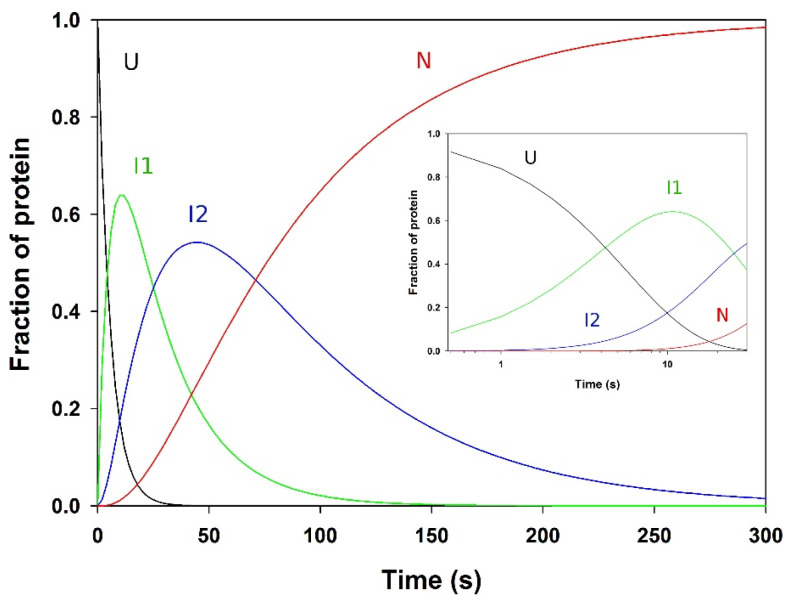
Simulation of the time-course of the fractional population of the various species for a four-state sequential model with two on-pathway intermediates (Scheme (5)). The simulation was performed using a program developed with the MATLAB computing platform (MathWorks Inc., Natick, MA, USA), with λ_1_ = 0.1754 s^−1^ (i.e., *τ*_1_ = 5.7 s), λ_2_ = 0.0412 s^−1^ (i.e., *τ*_2_ = 24 s) and λ_3_ = 0.0159 s^−1^ (i.e., *τ*_3_ = 63 s). The inset shows the first 30 s of the simulated kinetics, on a logarithmic time scale.

**Figure 10 biomolecules-11-01083-f010:**
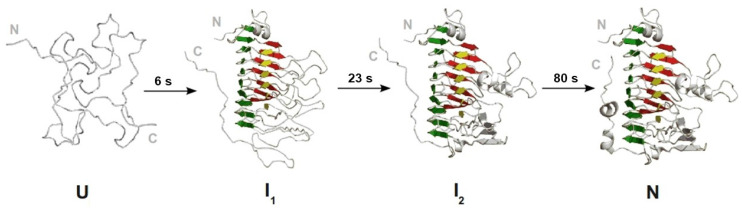
Representation of a possible folding pathway of PemA (pH 7, 25 °C). The sequential nature of the refolding process is indicated, and the two major intermediates are illustrated. The three β-sheets are in yellow (PB1), green (PB2), and red (PB3), and the *N*- and *C*-terminal ends of the polypeptide chain are indicated by N and C, respectively. Starting from a highly heterogeneous denatured state (U), the process of collapse (burst phase) and further minor structural rearrangements (phases 1 and 2) leads to a first intermediate species (I_1_), with native-like structure in the entire β-helix domain but unstructured peripheral loops (see text for details). This is followed by the folding of most of the peripheral loops and the formation of a second intermediate species (I_2_). Finally, the folding of the *C*-terminal loop and its docking against the parallel β-helix (PB2) (phase 4) leads to the final native structure. The values of the time constants for each step are indicated. Note that the various states in the model need to be described in terms of ensembles of conformers, and the experimental parameters are averaged over these ensembles.

**Table 1 biomolecules-11-01083-t001:** Kinetic parameters for the refolding of PemA after rapid mixing in 0.4 M GdmCl at pH 7, 25 °C.

Experiment	Burst Phase	Phase 1	Phase 2	Phase 3	Phase 4
Int. fluo.^ a^					
τ (s) ^e^	<0.003 ^g^	1.0 ± 0.3	5.7 ± 1.0	24 ± 8	63 ± 8
amplitude ^f^	0	−0.06 ± 0.01	0.20 ± 0.01	0.31 ± 0.02	0.55 ± 0.04
CD225nm ^b^					
τ (s) ^e^	<0.007	nd ^h^	5.1 ± 0.4	23 ± 4	63 ± 5
amplitude ^f^	0.56 ± 0.09	nd	0.25 ± 0.01	0.12 ± 0.02	0.07 ± 0.02
ANS fluo.^c^					
τ (s) ^e^	<0.003	0.14 ± 0.02	7.0 ± 1.0	25 ± 5	84 ± 10
amplitude ^f^	0.52 ± 0.02	0.032 ± 0.003	−0.13 ± 0.02	−0.15 ± 0.01	−0.10 ± 0.02
Acryl. ^d^					
τ (s) ^e^	<0.003	nd	7.0 ± 1.5	20 ± 1	115 ± 20
amplitude ^f^	−0.54 ± 0.05	nd	0.014 ± 0.006	0.10 ± 0.02	−0.018 ± 0.002

^a^ Intrinsic fluorescence. ^b^ Circular dichroism at 225 nm. ^c^ ANS fluorescence. ^d^ Quenching of fluorescence by acrylamide. ^e^ Time constant *τ* = 1/*k*, with *k* representing the macroscopic rate constants obtained using Equation (2). ^f^ Amplitude values are normalized as indicated in the text. The signs of the amplitudes derived from Equation (2) have been reversed, so that an increase of the measured parameter is denoted by a positive amplitude. ^g^ Lag phase. ^h^ nd, not detected. All values are the averages of results from three refolding experiments, and errors are calculated as standard deviations.

## Data Availability

Not applicable.

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
