# Peer review of "The Right-Handed Parallel β-Helix Topology of *Erwinia chrysanthemi* Pectin Methylesterase Is Intimately Associated with Both Sequential Folding and Resistance to High Pressure [Author-notes fn1-biomolecules-11-01083]"

_biomolecules, 2021, doi:10.3390/biom11081083_

Round 1
Reviewer 1 Report
This manuscript reports the folding mechanism of pectin methylesterase, a large multi-domain globular protein with the right-handed parallel beta-helix topology. The results contain interesting novel information about the folding behavior of this complex multi-domain protein, and hence, it deserves publication in biomolecules. However, this reviewer has a number of concerns about the presentation and the interpretations of the experimental data. The major concerns, listed below, may be considered by the authors before publication.
- The manuscript is too lengthy. It should be shortened extensively by eliminating unnecessary and speculative details. This reviewer suggests that the portions of the pressure-induced unfolding and the heat-induced unfolding may be completely omitted. These unfolding transitions are both irreversible, and hence the relationship with the reversible GdmCl-induced transition is not clear at all. The omission of these portions does not diminish the proposed folding mechanism of PemA and the quality of the manuscript. The title of the paper may also be changed accordingly. "Introduction" can be shortened by omitting the paragraph about phase P22 tailspike (from line 122 to line 136). This paragraph can simply omitted or move to "Discussion." "Discussion" can also be shortened by omitting speculative details.
- The interpretation about the lag phase is very confusing, and an extensive rewriting will be required. In page 12 and Table 1, the lag phase corresponds to phase 1 with its time constant of 1,0 s. In page 17, however, the formation of I2 (phase 3) shows a lag phase with a time constant of ca. 5 s because of the preceding accumulation of I1 (phase 2). The authors have proposed that this lag time (5 s) is in reasonable agreement with the experimentally observed lag time (ca. 5 s). This interpretation is very confusing, because the experimentally observed lag phase has a time constant of 1.0 s, and it occurred in phase 1 before phase 2. The transient species responsible for phase 1 is not identified in this manuscript. In page 20, the authors have again proposed that I1 reaches a maximum amount after ca. 9 s, a value which is in good agreement with that of the lag time (ca. 5 s) measured by tryptophan fluorescence experiments. Here this reviewer finds the same confusion.
- In relation to the above concern, the inset of Fig. 5A is shown in a rather wrong way. Here, the time axis is shown in a logarithmic scale. When we use the logarithmic time axis, even a simple single-exponential reaction curve shows a sigmoidal shape with a pseudo lag phase. The authors should not deceive the readers. The time axis should be shown in the normal scale as in all other figures.
- An additional concern about the lag phase (phase 1 in Table 1) is its absence in the folding kinetics measured by CD. If this lag phase represents the accumulation of an on-pathway folding intermediate, why is it unobservable by CD? The authors my provide a reasonable interpretation about the absence of phase 1 in the CD measurement
- In the legend of Fig. 3, the authors provide the equilibrium unfolding parameters of the GdmCl-induced unfolding transition measured by three different spectroscopic probes. This reviewer suggests that the authors may provide, somewhere in "Materials and Methods," an observation equation, which represents the relationship between the observed optical-parameter value and [GdmCl], used in the least-squares curve fitting analysis. People often cite Santoro & Bolen (1992) (Biochemistry 31, 4901) for this purpose.
- In page 22 (lines 857-859), the authors say as follows: the value of the folding rate constant (i.e. 0.8 s–1) extrapolated at 0 M GdmCl is about five order of magnitude lower than that (ca. 2·105 s–1) estimated on the basis of protein contact order. Here, which contact order parameter, the original relative contact order or the absolute contact order, was used to predict the folding rate constant? It is now well established that the relative contact order cannot be used to predict the rate constant of the proteins that accumulate the folding intermediate, and the chain length and other structure-based parameters show a good correlation with the folding rate of non-two-state proteins (see Galzitskaya et al (2003) Proteins 51, 162; Kuwajima (2020) Biomolecules 10, 407).
- There are the following minor typographic errors:
Lines 113-114: pectin methylesterase-->PemA
Line 122: P22 tailspike endorhamnosidase-->TSP
Line 280: (see pressure-induced unfolding-->(see pressure-induced unfolding)
Reviewer 2 Report
In this manuscript, the dynamics of protein folding has been analyzed in detail using spectroscopic techniques. Several methods and their mathematical analysis clearly show that this folding process can be divided into four phases. Numerous references are cited, and it is also discussed in depth that this kinetic analysis agrees well with the results and discussion of previous studies.
I think it is also reasonable to assume that the two phases between the first and the last phase are derived from metastable "two obligatory, on-pathway, intermediates". However, since these discussions are based on measurements of the environment of the amino acid side chains of each part, the conformation of the main chain cannot be clearly determined. Therefore, the models of I1 and I2 shown in Figure 10 seem to me to be over speculation.
It was shown that some tryptophan side chains are buried in the intermediate structure, but it is unclear which tryptophan side chains. It is also unclear to me, based on the results and the previous studies presented by the authors, whether the β-sheets present in the known native structure are formed in the intermediate I1 and I2, or whether some other transient metastable secondary structure is formed. The evidence that the three β-sheets are already formed in I1 must be more clearly shown. For example, it would be desirable to show, by high-pressure NMR or other appropriate method, that the three β-sheets are indeed formed in I1. Alternatively, the high-pressure far-UV CD results should be further analyzed to see if the β-sheets content under high-pressure conditions corresponding to I1 matches the β-sheets content calculated from possible I1 model. It is also useful to follow the changes in dynamics using point mutation.
Although the quality of the experimental data and mathematical analysis is sufficient and the interpretation of the results seems reasonable, I do not believe that sufficient evidence has been provided for the construction of the model in Figure 10. If the authors want to depict the conformation of the polypeptide chain beyond the schematic model, they should present the more data from other structural analyses or provide the theory or software used to construct the model not in the acknowledgements section but in the method section.
miner point
Line 442: Either the description of heat-induced unfolding should be omitted or the data should be presented.
Reviewer 3 Report
The paper by J. Guillerm et al. is an elegant investigation of the unfolding and refolding mechanisms of a large tandem-repeat protein. They chose the pectin methylesterase from Erwinia chrysantemi as their model and their observations corroborate the occurrence of a two-state mechanism for unfolding, induced by either chemical agents, temperature or pressure. On the other hand, they evidenced a more complex refolding mechanism, with the occurrence of at least two sequential partially folded intermediates. In my opinion, the structural data are very exhaustive and well presented and no significant change is needed before publication. I do have two concerns, that I believe the authors should address in a revised version of the manuscript:
1) In the methods and discussion sections they mention a colorimetric assay with bromocresol purple to assess enzymatic function. I could not find in the results any part where this assay is performed and discussed.
2) I believe the authors should double check the procedure they reported for the cloning of the vector: cloning the protein coding sequence between NdeI and XhoI in the pET20b+ vector would have removed the pelB signal sequence, leading to cytoplasmic protein accumulation, which would not be compatible with the osmotic shock strategy used for periplasmic protein extraction. They should carefully control this part.
Round 2
Reviewer 1 Report
The manuscript has been revised appropriately, and it can be published now.